# Fast Training of Sparse Graph Neural Networks on Dense Hardware

## Abstract

Graph neural networks have become increasingly popular in recent years due to their ability to naturally encode relational input data and their ability to operate on large graphs by using a sparse representation of graph adjacency matrices. As we look to scale up these models using custom hardware, a natural assumption would be that we need hardware tailored to sparse operations and/or dynamic control flow. In this work, we question this assumption by scaling up sparse graph neural networks using a platform targeted at dense computation on fixed-size data. Drawing inspiration from optimization of numerical algorithms on sparse matrices, we develop techniques that enable training the sparse graph neural network model from Allamanis et al. (2018) in 13 minutes using a 512-core TPUv2 Pod, whereas the original training takes almost a day.

## 1 Introduction

As we seek to apply high capacity neural network models to new applications, we often encounter data that is not structured as a vector, image, sequence, or set. In these cases, graph neural networks (GNNs) (Scarselli et al., 2009; Li et al., 2016) can be appealing because they operate on graph-structured inputs. Each node can have data (e.g., an image) associated with it, and edges can encode different kinds of relationships between nodes. The forward pass of a GNN model can be interpreted as nodes exchanging messages with each other along edges of a graph (Gilmer et al., 2017), combining the local per-node information with information about the surrounding context in a flexible manner. Domains where GNNs have found success include chemistry (where graph-structured representations of molecules are mapped to predictions about molecular properties) and program analysis (where a graph-structured representation of a program encodes a combination of syntactic information and semantic relationships between program entities).

In the program analysis application of Allamanis et al. (2018), graphs have up to 20,000 nodes. Computing with dense $20,000 \times 20,000$ adjacency matrices is expensive, so common practice is to operate on sparse adjacency matrices. With sparse representations it is straightforward to operate on graphs with $\approx 100,000$ nodes per batch. The model can be implemented efficiently using sparse operations such as scatter and gather, or `sparse_tensor_dense_matmul` in TensorFlow.

From a hardware perspective, modern advances in deep learning are partly due to GPUs (LeCun et al., 2015). Modern GPUs have many cores, and the threads started by a single program can access memory, branch, and terminate independently from other threads. Even though performance is maximized when threads do not diverge (facilitating efficient dense linear algebra operations), relatively fast implementations of sparse operations are possible (Bell & Garland, 2009). New "AI accelerators" such as Google's Tensor Processing Units (TPU), Intel's Nervana Neural Network Processor, and Nvidia's Volta architecture use domain specific architectures (DSAs) (Hennessy & Patterson, 2019) to perform matrix multiplication. We refer to these collectively as "dense hardware". While it may seem hopeless to rewrite sparse GNN computations in terms of dense matrix multiplies that are needed to access dense hardware computational bandwidth, we show that it can be done efficiently enough so that training on a single dense device is comparably fast to training the sparse model on a GPU, and that scaling out to more cores yields *the fastest wall-clock training time by orders of magnitude* for a canonical sparse GNN model (to the best of our knowledge).

Our approach will be to think of appropriately-sized matrix multiplication as a primitive in a "virtual machine". The problem then becomes "compiling" the GNN propagation on a sparse graph to

only use primitives in the virtual machine. The first key observation is that if we can reveal a low bandwidth structure (Díaz et al., 2002) in the graph adjacency matrix, then GNN propagation can be losslessly expressed in terms of three applications of a dense batched matrix multiply primitive. This reduces the cost of a step of densified GNN propagation from $O(N^2H)$ to $O(NBH)$, where $N$ is the number of nodes, $B$ the bandwidth, and $H$ the node embedding dimension. The second key observation is that it is possible to permute nodes in the graphs from Allamanis et al. (2018)—which we take to be representative of the types of graphs that arise in program analysis tasks—to expose approximately low bandwidth structure using algorithms from sparse numerical computing.

After applying bandwidth reduction and implementing GNN message propagation for low bandwidth graphs, we achieve performance on TPUv2 hardware that is already competitive with a highly optimized sparse GPU implementation. We then take advantage of the ease of scaling out TPU computations to many cores and explore large batch training. (We are not aware of successes scaling out to equivalent size GPU clusters and believe it to be far less trivial; see the discussion of Ma et al. (2018) in Section 6.) We are able to achieve near-linear scaling as the number of TPU cores grows up to 128 and then see some diminishing returns beyond that, confirming the pattern observed by Shallue et al. (2018). On the challenging *VarMisuse* problem from Allamanis et al. (2018) (single-GPU training takes close to a day), we reach near state-of-the-art accuracy in 13 minutes.

In summary, the contributions of this paper are as follows:

1. identifying approximate low bandwidth as a structure that enables efficient dense implementation of GNN propagation while also being present in real-world program analysis data;

2. applying algorithms from sparse numerical computing to reduce bandwidth for GNN graphs, and developing a fast algorithm for dense propagation in sparse low bandwidth GNNs;

3. empirical results on large-batch training for GNNs, extending the large-batch steps to validation accuracy study of Shallue et al. (2018) to sparse GNN;

4. empirical results achieving orders of magnitude faster time-to-accuracy on the dataset and task from Allamanis et al. (2018).

More broadly, the recipe of compiling sparse graph structures to dense operations could be applied to other families of graphs (examples are mentioned in the Discussion), and since we observe that GNN training is robust to dropping some non-conforming (out of bandwidth) edges during training time, even approximate sparse structure can be leveraged. Finally, given that we achieve near state-of-the-art accuracy with a densely-trained model, our work raises questions about whether there are alternatives to sparse GNNs that would achieve even better accuracy at the same computation cost.

## 2 BACKGROUND

**Graph neural networks** There have been many developments and applications of GNNs over recent years (Zhou et al., 2018; Wu et al., 2019). Of particular relevance are Scarselli et al. (2009), which presented the original GNN model, and Li et al. (2016), which developed the gated GNN (GGNN) variant used by Allamanis et al. (2018) and thus also by us.

A GGNN encoder takes as input a graph on $N$ nodes with initial node embeddings $E^{(0)} \in \mathbb{R}^{N \times H}$, and after a fixed number $T$ of time steps produces final node embeddings $E^{(T)} \in \mathbb{R}^{N \times H}$ that combine local and neighborhood information. In each time step, each node computes a message from its current embedding using a learnable linear map parametrized by $W \in \mathbb{R}^{H \times H}$, and broadcasts the message to all its neighbors. Each node sums up its received messages, and updates its embedding using a GRU cell (Cho et al., 2014). Writing $A \in \{0, 1\}^{N \times N}$ for the (transposed) adjacency matrix ($a_{ij} = 1$ if there is an edge from $j$ to $i$), the forward pass of a GGNN encoder is captured by

$$E^{(t+1)} = \mathrm{GRU}(AE^{(t)}W, E^{(t)}) \quad \text{for } t = 0, 1, \ldots, T-1. \tag{1}$$

When edges are of $P$ discrete types, separate weights $W_p$ parametrize the map from embeddings to messages for each edge type: $E^{(t+1)} = \mathrm{GRU}(\sum_{p=1}^{P} A_p E^{(t)} W_p, E^{(t)})$, where $A_p$ only contains edges of type $p$. The pre-multiplications by the sparse matrix $A$ (or $A_p$) are the crucial sparse operations that we seek to replace with fast primitives on dense hardware. The final node embeddings

$E^{(T)}$ can be used directly (e.g., for node classification), pooled together into an embedding of the whole graph, or, as in our case, fed into an output layer that for each graph selects a node.

**Program graphs**  Program source code can be represented as a string and treated as text, but this ignores its precisely defined structure (the syntax of the programming language) and semantics. To improve performance, Allamanis et al. (2018) proposed to represent source code as a graph. The "backbone" of the graph is the code's abstract syntax tree (AST). Additional edges of different types, capturing the syntax and semantics of the language, are added into the graph. For example, there is an edge type linking AST nodes to their children, an edge type linking each token in the source code to the next one, and an edge type linking a variable token to the variables it is computed from. Given source code, the program graph can be pre-computed using static analysis.

**Variable misuse**  Introduced by Allamanis et al. (2018), *VarMisuse* is the following task: Given a snippet of code with one of the usages of a variable masked out (the *hole*) select which variable is used in the hole from a set of candidates. Once trained, the model can be run on test code by masking out variable usages and finding where model predictions disagree with the actual code.

The model trained by Allamanis et al. (2018) is a neural network consisting of a GGNN encoder, and a readout layer that applies a learned linear map $\mathbb{R}^H \to \mathbb{R}$ to the candidate node embeddings to obtain per-node logits, so that a softmax yields the normalized probability of each candidate node filling the hole. Allamanis et al. (2018) report finding bugs in open source projects using this model.

**Sparse batching**  Instances in a training set of (program) graphs have differing node counts, so the question of efficient batching arises. With sparse adjacency representations of graphs an elegant batching scheme is possible: multiple training graphs can be packed into a single *supergraph* of fixed maximum size, until no more graphs fit (Allamanis et al., 2018). Individual training graphs are represented by disjoint connected components in this supergraph, and since message passing only proceeds along graph edges, different training graphs do not affect each other. Thanks to the sparse representation, no quadratic overhead is incurred by the large number of nodes in the supergraph.

**Sparse numerical linear algebra**  Performing sparse matrix multiplication on modern hardware can be difficult due to the irregular memory access patterns. A large body of work has studied implementing sparse matrix operations on a variety of architectures (Saad, 2003, Sec. 3). One consideration is the storage scheme of the matrix, another the ordering of rows and columns, which can significantly affect the performance of certain operations, see, e.g., Tinney & Walker (1967).

When the matrix is a graph adjacency matrix, this ordering corresponds to an ordering of nodes. Finding an ordering of graph nodes to optimize an objective is known as a *graph layout problem* (Díaz et al., 2002). Of particular interest in this work is *bandwidth minimization*. The *bandwidth* of a square matrix $A = \{a_{ij}\}_{N \times N}$ is the smallest $B \in \mathbb{N}_0$ such that $a_{ij} = 0$ whenever $|i - j| > B$. This is an NP-hard problem, but a popular and fast (linear-time) heuristic algorithm is Reverse Cuthill McKee (RCMK) (Cuthill & McKee, 1969).

## 3  METHOD

As an overview, the full training pipeline consists of the following components:

1. *Compilation*: a one-time preprocessing step that "compiles" each training point by finding an efficient computation schedule for it. In our case this amounts to finding a permutation of nodes in each training graph such that the resulting adjacency matrix has low bandwidth.

2. *Input pipeline* (not a novel contribution): efficient reading of compiled training data from disk, assembly of *supergraphs* via *sparse batching* (see Section 2), and feeding them to training hardware so that it is not blocked on waiting for training data. When training on multiple cores, each core receives its own supergraph.

3. *Model*: an algorithm that executes the GGNN encoder logic by invoking operations that are fast on the target hardware. In our case, this corresponds to three invocations of batch matrix multiplication, which together perform the message passing logic for low-bandwidth graphs.

### 3.1 COMPILATION: REDUCING BANDWIDTH

Given a sparse graph adjacency matrix, we first permute the nodes to expose reduced bandwidth structure using the RCMK algorithm. Since predictions from a GNN are equivariant to node permutation, we simply permute the labels as we permute the nodes in the graph. Compiling a training graph $(V, E)$ takes linear time $\mathcal{O}(|V| + |E|)$, and this is done as a one-time preprocessing step on each training graph in the dataset.

### 3.2 MODEL

Our implementation consists of two parts: 1) an algorithm that expresses the graph message passing operation on low-bandwidth graphs in terms of batch matrix multiplications, and 2) an efficient implementation of this algorithm with a memory layout that avoids tensor transpositions.

**Efficient densification of reduced bandwidth GNN propagation**  A general class of efficient operations on dense hardware can be expressed using the `einsum` operation (SciPy, 2019), a particular case of which is *batch matrix multiplication* (BMM): Given two tensors of compatible shapes $[d_1, \ldots, d_{n-2}, m, k]$ and $[d_1, \ldots, d_{n-2}, k, n]$, it performs the $d_1 \cdots d_{n-2}$ matrix multiplications in the last two dimensions, yielding a tensor of shape $[d_1, \ldots, d_{n-2}, m, n]$.

As a first step, suppose that $A$ is a square, block diagonal matrix with $K$ equal block sizes $S$, i.e. with overall shape $KS \times KS$. Writing $C_1, \ldots, C_K$ for the individual non-zero $S \times S$ blocks of $A$, the matrix product of $A$ with another matrix $E$ of compatible shape $KS \times H$ can be expressed as

$$
AE = \begin{bmatrix} C_1 & & \\ & \ddots & \\ & & C_K \end{bmatrix} \begin{bmatrix} E_1 \\ \vdots \\ E_K \end{bmatrix} = \begin{bmatrix} C_1 E_1 \\ \vdots \\ C_K E_K \end{bmatrix} \tag{2}
$$

where the $E_k$'s are $S \times H$ blocks of $E$. Thus it is possible to only store the non-zero blocks $C_1, \ldots, C_K$ (instead of the entire matrix $A$) and to implement the matrix product $AE$ as

```
A_E = reshape(BMM([C_1,...,C_K], reshape(E, [K,S,H])), [KS,H]).
```

This idea can be extended to the low-bandwidth case by employing two additional BMMs to "fill the gaps" around the block corners in the block diagonal case (see right). Formally, if $A$ is a $KS \times KS$ matrix with bandwidth $B \leq S - 1$, then $AE =$ 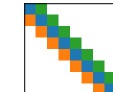

$$
\begin{bmatrix} C_1 & U_2 & & & \\ L_1 & C_2 & U_3 & & \\ & \ddots & \ddots & \ddots & \\ & & L_{k-2} & C_{k-1} & U_k \\ & & & L_{k-1} & C_k \end{bmatrix} \begin{bmatrix} E_1 \\ E_2 \\ \vdots \\ E_{k-1} \\ E_k \end{bmatrix} = \begin{bmatrix} C_1 E_1 \\ C_2 E_2 \\ \vdots \\ \\ C_k E_k \end{bmatrix} + \begin{bmatrix} U_2 E_2 \\ U_3 E_3 \\ \vdots \\ U_k E_k \\ 0 \end{bmatrix} + \begin{bmatrix} 0 \\ L_1 E_1 \\ L_2 E_2 \\ \vdots \\ L_{k-1} E_{k-1} \end{bmatrix} \tag{3}
$$

where the blocks $C_k, U_k, L_k$, are all $S \times S$ matrices, and additionally the $U_k$s are strictly *lower*-triangular and the $L_k$s are strictly *upper*-triangular. The three matrices on the right-hand side can be computed using batch matrix multiplications, see Algorithm 1 for details.

In each training run, we choose the block size $S$ depending on the bandwidths $B$ that we wish to handle. We still apply our low-bandwidth message passing to graphs that violate the assumption $B \leq S-1$, but some edges will be ignored (messages are not passed along them). Specifically, edges $(i, j)$ with $|i - j| < S$ are always included, those with $S \leq |i - j| < 2S$ are sometimes included, and those with $|i - j| \geq 2S$ are always ignored (see illustration above Equation 3). Interestingly, we observe that dropping these edges at training time (but including them at test time) still achieves near state-of-the-art accuracy.

**Memory layout**  To efficiently feed data into the dense hardware, the data corresponding to each matrix in the batch matrix multiplies need to be contiguous in memory. When tensor memory is stored in lexicographic order, this corresponds to those dimensions being the last in the respective tensors. Thus, the BMM operation described above is more efficient than an `einsum` over the same values but with the order of dimensions shuffled. If memory needs to be rearranged before applying the BMM operation, this incurs overhead. To some extent, these concerns can be hidden by a compiler, but we found it beneficial to explicitly choose the memory layout.

---

**Algorithm 1** Low-bandwidth GGNN Message Propagation Step

---

**Input:** tensors of the following shapes:
- `Cks [K, S*P, S]` ($C_k$'s concatenated across the $P$ edge types)
- `Uks [K-1, S*P, S]` ($U_k$'s concatenated across the $P$ edge types)
- `Lks [K-1, S*P, S]` ($L_k$'s concatenated across the $P$ edge types)
- `Eks [K, S, H]` ($E_k$'s; reshaped tensor of node embeddings)
- `Wps [P*H, H]` ($W_p$'s concatenated edge transform matrices for $P$ edge types)

**Output:** `new_node_embeddings [K, S, H]` ($E_k^{(t+1)}$'s; reshaped new node embeddings)
   {First compute `[K, S*P, H]` tensor `pre_messages` storing the sum of node embeddings (`H`) sent to each node (`K*S`) via an edge of each type (`P`), before transforming by edge model.}

```
1:  pre_messages            = einsum(kzs,ksh->kzh, Cks, Eks)
2:  pre_messages[0:K-1] += einsum(kzs,ksh->kzh, Uks, Eks[1:K])
3:  pre_messages[1:K]   += einsum(kzs,ksh->kzh, Lks, Eks[0:K-1])
4:  pre_messages = reshape(pre_messages, [K, S, P*H])
5:  incoming_messages = einsum(ksy,yh->ksh, pre_messages, Wps)
6:  new_node_embeddings = GRU_cell(incoming_messages, Eks)
```

---

The challenge in the GGNN model is that we need to perform two kinds of matrix-multiplies: 1) multiplication by the adjacency matrix to aggregate messages incoming to each node, and 2) multiplication by edge weight matrices to transform node embeddings into messages (see Equation 1). In Algorithm 1, we lay out memory and perform a message passing step in a way that keeps the dimensions being multiplied in the last dimensions, and avoids any tensor transpositions. TensorFlow code for our model is provided with the submission.

## 4   DATA

The dataset released by Allamanis et al. (2018) consists of *VarMisuse* instances constructed from source code of 25 open source C# projects. Each program graph contains a special *hole node*, representing the location in code for which a variable is to be predicted, and a set of *candidate nodes*, representing all type-correct in-scope variables that could potentially fill the hole. The identity of the correct candidate is also provided, so this is a supervised learning classification task.

The data comes with two pre-defined train-validation-test splits: *SeenProj*, where validation and test data come from the same projects as the training data, and *UnseenProj*, where they come from disjoint projects. Allamanis et al. (2018) provide more detailed results using *SeenProj*, so to allow comparison we report results using the provided *SeenProj* splits. As a one-time pre-processing step we discarded nodes from all graphs that cannot affect model predictions (and so do not influence training) due to messages from these nodes not being able to reach the candidate nodes (see Appendix E for details). This tweak gave a small speed-up to all methods; without it the results were similar and the conclusions identical. There are 124,592 training graphs in the dataset. The largest number of nodes in a single graph is 18,582. Preserving the provided ordering of the non-eliminated nodes, the largest ("original") bandwidth of a graph adjacency matrix would be 18,535. See also the blue and orange histograms in Figure 1 (left). There are $P = 22$ edge types.

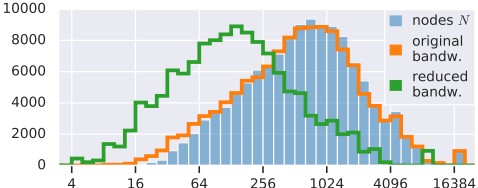

| bandwidth bound $B$ | training data % |
|---|---|
| 128 | 48.9% |
| 256 | 69.2% |
| 512 | 83.5% |
| 1024 | 91.9% |

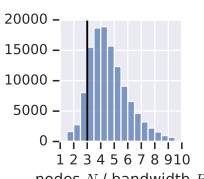

Figure 1:   Statistics of the training graphs. Left: node counts and adjacency matrix bandwidths, horizontal axis is log-scaled. Middle: proportion of graphs whose bandwidths can be reduced below the given threshold $B$. Right: ratios between node count and adjacency matrix bandwidth after applying bandwidth reduction.

## 5 EXPERIMENTS

We report results from four sets of experiments, demonstrating the following findings:

1. Program graphs often exhibit low-bandwidth structure (Section 5.1). For example, more than $80\%$ of all training graphs in the (Allamanis et al., 2018) dataset are representable by an adjacency matrix of bandwidth lower than $512$.

2. Our GNN propagation method for low-bandwidth graphs allows fast processing of training data (Section 5.2). A single TPUv2 device achieves similar speed as a highly optimized sparse implementation on GPU, and the speed scales almost linearly with more cores.

3. Large-batch training allows training the model to the same validation accuracy significantly faster (Section 5.3), and this is true despite ignoring some edges in large-bandwidth graphs. Training on a 512-core TPUv2 slice can reach the target in $13$ minutes.

4. Instead of using all training points, filtering out "troublesome" graphs can be considered: we can drop graphs with large bandwidths and use our method losslessly (not ignoring any edges), or we can drop graphs with too many nodes, so that full densification becomes feasible for the remaining (smaller) graphs. We find that these approaches scale less well or lead to lower validation accuracies. For space, details are left to Appendix C.

### 5.1 ALLAMANIS ET AL. [2018] DATA HAS LOW BANDWIDTH

Figure 1 (left and middle) show that a compilation step can be designed such that most program graphs in the Allamanis et al. (2018) dataset are reduced to a low-bandwidth representation. Moreover, Figure 1 (right) shows that after our compilation step using the RCMK algorithm, the bandwidth $B$ of virtually all training graphs in the dataset is significantly smaller than their number of nodes $N$. The ratio of 3 is highlighted as it represents the crossover point at which the theoretical $N^2$ cost of full densification surpasses the $3NB$ cost of covering the bandwidth using our three BMMs.

### 5.2 TRAINING SPEED IN TRAINING GRAPHS PROCESSED PER SECOND

First, as a baseline we reimplemented the GGNN model of Allamanis et al. (2018), for which they reported a training speed of 55 graphs/second on a TitanX GPU, using a hidden dimension $H = 64$. We optimized our input pipeline for speed, and using a newer V100 GPU we achieved a training speed of 80 graphs/second, despite using double the hidden dimension $H = 128$.

We also implemented our low-bandwidth model using different block sizes $S$. A larger block size covers more edges in the (large-bandwidth) training graphs, but the memory cost of storing the wider densified block diagonals decreases the number of nodes in the supergraph that we can fit into memory. A more detailed analysis appears in Appendix A; the table in Figure 2 (left) shows the overall training speeds achieved in training graphs processed per second.

The GPU model does not have a block size parameter, so to obtain the corresponding speeds we dropped edges lying outside of the corresponding bandwidth. However, unlike the low-bandwidth model, the GPU model did not visibly benefit from decreasing the block size (the benefit of not having to pass messages along a few edges far from the diagonal is negligible).

| graphs/sec | GPU model | Low-bandwidth model | | | | |
|---|---|---|---|---|---|---|
| | | number of TPUv2 cores trained on | | | | |
| Block size $S$ | trained on V100 | 2 (16GB) | 8 (device) | 32 | 128 | 512 |
| all edges | 80 | - | - | - | - | - |
| 1024 | 80 | 25 | 100 | 400 | 1,600 | 4,500 |
| 512 | 80 | 75 | 260 | 1,000 | 3,900 | 11,000 |
| 256 | 80 | 110 | 450 | 1,800 | 6,900 | 21,000 |
| 128 | 80 | 160 | 630 | 2,500 | 9,700 | 24,000 |

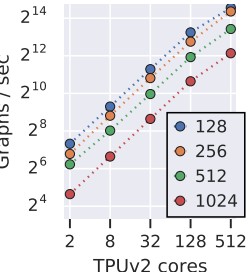

Figure 2: Training speed (graphs/s) on the full training data (ignoring non-conforming edges).

A single TPUv2 device has $8$ cores with 8GB of RAM each, so for comparison we also estimated the speed on a hypothetical TPUv2 slice with just two cores, matching the 16GB of RAM available on a V100 GPU. As the block size decreases from $512$ to $256$, this 16GB slice surpasses the GPU training speed. A single TPUv2 device as well as TPUv2 slices with more cores lead to further improvements in the training speed: Figure 2 (right) visualizes the almost ideal linear scaling.

### 5.3 Training time to validation accuracy

As explained in Section 4, we evaluate our approach on the *SeenProj VarMisuse* task of Allamanis et al. (2018). They report a test set accuracy of $85.5\%$ for this task, however 1) their model is trained on a larger dataset than the one released, and 2) it includes an additional learnable component (node label embeddings). Using the information provided by Allamanis et al. (2018) and our finding that accuracies on the validation set lie 5 percentage points lower than on the test set (see Figure 4 in Appendix B), we set the target validation accuracy at $78\%$. See Appendix B for the full justification.

We trained the GPU model and our low-bandwidth model capturing different bandwidths. Scaling up to larger TPU slices we were able to train the latter using large batch sizes, and following Shallue et al. (2018) we performed a fresh learning hyper-parameter sweep for each batch size. All runs used the SGD optimizer with momentum; see Appendix E for hyper-parameter ranges.

For both the sparse GPU model and our low-bandwidth model the parameters learned during training are the same: the per-edge-type matrices $W_p$ (including a bias term), the weights of the GRU cell, and the output layer. This means that at evaluation/prediction time, the weights of a low-bandwidth model trained with any block size $S$ can be used by the sparse GPU model, so that for evaluation messages are passed across all edges in the validation graphs. We found that this performed better than only passing messages along edges within the same block size as the one used for training.

For each training run, we evaluated training checkpoints at regular intervals on the full validation data and smoothed the resulting curves. We then found the earliest checkpoint at which this validation accuracy curve exceeded the $78\%$ target. Figure 3 reports the results. The visualization on the right extends the study of Shallue et al. (2018) to the sparse GGNN model.

| Block size $S$ | GPU model trained on V100 | Low-bandwidth model | | | |
|---|---|---|---|---|---|
| | | number of TPUv2 cores trained on | | | |
| | | 8 | 32 | 128 | 512 |
| all edges | 75,000 (19 h) | - | - | - | - |
| 1024 | 145,000 (32 h) | 56,000 (17 h) | 8,800 (2 h) | 2,600 (34 min) | 1,200 (22 min) |
| 512 | 180,000 (50 h) | 18,000 (4 h) | 3,000 (40 min) | 900 (17 min) | 700 (13 min) |
| 256 | 230,000 (52 h) | 38,000 (7 h) | 3,000 (35 min) | 1,800 (22 min) | 1,000 (18 min) |
| 128 | not reached | not reached | not reached | 8,000 (75 min) | not reached |

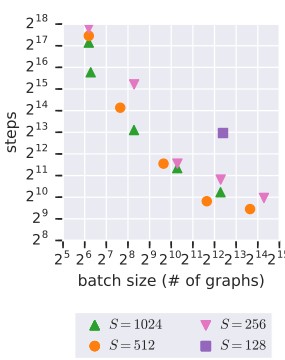

Figure 3: Training steps (and training time) until 78% validation accuracy.

## 6 Related work

**Sparse linear algebra**   Our approach to sparse matrix multiplication is similar to approaches used in classical sparse linear algebra. On vector processors a common storage scheme for sparse matrices is the diagonal representation where a matrix is represented as a vector of integers corresponding to the non-zero diagonals along with a dense matrix containing the entries on these diagonals (Saad, 2003, Sec. 3.4). Our storage scheme can be considered a block-diagonal version of this storage scheme that is more efficient to process using the systolic array structure on TPUs.

**Block-sparse neural networks**   In previous work authors have explored block-sparse connectivity patterns as ways of combining the benefits of sparsity with the performance of dense matrix multiplication on modern hardware. Graph neural networks can be interpreted as sparse neural networks

with a fixed binary weight matrix (the adjacency matrix) and a large hidden state (the node states). Similarly, *grouped convolutions* (Xie et al., 2017) and *depth-wise separable convolutions* (Simonyan & Zisserman, 2015) can be interpreted as block-sparse operations. Narang et al. (2017) looked into learning block-sparse weight matrices for RNNs. Gray et al. (2017) explore the implementation of highly performant CUDA kernels for block-sparse matrix multiplication.

**Speeding up neural network training**  Goyal et al. (2017) used large-batch training to train a ResNet-50 model on ImageNet in one hour. A subsequent chain of work has further reduced the training time to minutes (Akiba et al., 2017; You et al., 2018; Jia et al., 2018). Our work can be seen as an analogous first step for the GNN model and task from Allamanis et al. (2018). Shallue et al. (2018) recently performed an extensive empirical investigation of large-batch training on a variety of (non-sparse) models on multiple image classification and language modeling tasks.

There have been some efforts to speed up GNN training using GPUs. Deep Graph Library (Wang et al., 2019) is a framework for training a variety of neural network models that involve passing messages in a sparse graph. The main optimizations are sparse batching (discussed in Section 2), batching together message computations across nodes, and fusing message and node update functions (e.g., using a sparse-dense matrix multiply). Our GPU baseline uses all these optimizations and an additional optimization from Allamanis et al. (2018) where node states can be pre-aggregated by (receiving node, edge type) pairs when using linear edge transformations. Ma et al. (2018) also develop a framework for efficient GNN training, using a number of lower-level optimizations including custom GPU kernels for scatter and gather operations, and a ring-based data-loading pipeline to reduce contention when transferring data from the host CPU to multiple GPUs. They show that the custom GPU kernels lead to runtimes that range from 50% to 90% that of a baseline TensorFlow implementation, and the ring-based data loading allows near-linear scaling up to 8 GPUs. Scaling to more than 8 GPUs seems non-trivial (and is not reported).

Our focus is on problems where there are many graphs of large size (up to $\approx 100{,}000$ nodes in a graph). Alternative approaches are available in the domain of a single very large graph like is found in social networks or knowledge graphs. In these cases, methods like GraphSAGE (Hamilton et al., 2017) or the model from Dai et al. (2018) are likely more appropriate.

# 7 DISCUSSION AND FUTURE WORK

At the outset of this work, it was unclear if dense hardware could be made competitive with GPUs for training sparse graph neural networks. The main result of this paper is to show that for a prominent application of sparse graph neural networks on relatively large graphs, it is indeed possible. Moreover, we studied large batch training for graph neural networks and showed that it is possible to significantly speed up time-to-accuracy by scaling out to many cores and increasing batch sizes.

More generally, we have presented an example of a recipe for tackling the problem of sparse graph message propagation on dense hardware. The general recipe is to think in terms of an abstract virtual machine, comprising a *runtime* consisting of operations that are fast on dense hardware and a *compiler* that rewrites the graph propagation step in 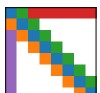 terms of runtime primitives. It would be straightforward to add more runtime primitives; we mention two that we are particularly excited about. The first is "supernode" propagation where some nodes are allowed to communicate with all nodes, which could be implemented by taking dense slices from the first rows and columns of the adjacency matrix (see right). The second is a general block-sparse matrix multiply like in the CUDA primitives of Gray et al. (2017). The compilation step for these primitives will be an interesting combinatorial optimization problem. Finally, it would be interesting to identify other primitives to handle specific classes of graphs (e.g., small world networks).

In this work, we held fixed the graph structures that were provided by Allamanis et al. (2018), and already there is low bandwidth structure that can be leveraged for efficient computation. However, going forward we would like to re-design graph structures that perform well but are constrained to support even faster computation (e.g., can we design a bandwidth 128 model that performs equally well?) We are also eager to collect much larger datasets and study how performance scales with more data. The techniques presented in this paper make this feasible.

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

APPENDIX FOR ICLR 2020 SUBMISSION: FAST TRAINING OF SPARSE GRAPH
NEURAL NETWORKS ON DENSE HARDWARE

## A    TRAINING SPEED ANALYSIS

In Section 5.2 we saw that a smaller block size $S$ allowed processing a larger number of training
graphs per second. From a computational cost perspective, there are two forces at play as the block
size $S$ decreases: 1) the $S$ dimension of the densified block diagonals decreases, which 2) in turn
allows fitting supergraphs with more nodes $N$ into memory. As the cost of the two types of matrix
multiplies in our low-bandwidth GGNN message passing algorithm (Algorithm 1) is $O(NSH)$
and $O(NH^2)$, respectively, a priori it is not clear how these two forces contribute to the increased
training speed. The purpose of this section is to shed more light on this.

### A.1    NUMBER OF NODES IN A SUPERGRAPH

Having fixed the hidden dimension $H = 128$ and the number of $T = 8$ steps of GGNN message
passing in all our experiments, we can find the (approximate) largest number of nodes in a *super-
graph* that still fit into memory, given a model and hardware combination. All models need to store
the $(T + 1) \times N \times H$ node embeddings $E^{(t)}$ for $t = 0, 1, \ldots, T$. The memory cost of the sparse ad-
jacency representation on a GPU is negligible, and we can train on *supergraphs* with up to 110,000
nodes on a V100 GPU with 16GB of RAM, which on average translates to 73 training graphs per
*supergraph*. Our low-bandwidth model requires storage of three densified block diagonals of shapes
roughly $[N, S]$, which starts to significantly add to the memory cost for larger block sizes $S$. Ta-
ble 1 shows the limits for this model, depending on the chosen block size $S$. As expected, covering
thinner bands allows fitting larger *supergraphs* (packing more training graphs) into memory.

Table 1:  Per-core *supergraph* size, when training our low-bandwidth model on TPUv2.

| Block size $S$ | 128 | 256 | 512 | 1024 |
|---|---|---|---|---|
| Number of nodes $N$ in a *supergraph* | 70,272 | 65,536 | 49,125 | 28,672 |
| Average number of graphs in a *supergraph* | 42 | 39 | 25 | 10 |

Recall that each TPUv2 core receives its own supergraph, so the actual batch size will scale linearly
with the number of TPUv2 cores trained on.

### A.2    NUMBER OF GRADIENT STEPS PER SECOND

Decreasing the block size $S$ allows fitting more nodes $N$ into memory, so a priori it is not clear
how the wall time to execute a single step of graph message passing would be affected. Table 2
shows the number of global steps (gradient updates) per second. Reading along columns of the
table, we can observe that performing a single gradient update with a block size $S = 512$ is slightly
slower than using a block size $S = 1024$, but as the block size $S$ decreases further, not only does a
single supegraph fit more nodes (and therefore, more training graphs), but also each gradient update

| step/sec | GPU model | Low-bandwidth model | | | |
|---|---|---|---|---|---|
| Block size | trained | TPUv2 cores trained on | | | |
| $S$ | on V100 | 8 | 32 | 128 | 512 |
| all edges | 1.05 | - | - | - | - |
| 1024 | 1.05 | 1.34 | 1.29 | 1.26 | 0.90 |
| 512 | 1.05 | 1.30 | 1.27 | 1.20 | 0.88 |
| 256 | 1.05 | 1.46 | 1.44 | 1.40 | 1.07 |
| 128 | 1.05 | 1.89 | 1.87 | 1.82 | 1.12 |

Table 2:  Global steps (gradient updates) per second for different model, hardware, and block size combina-
tions.

gets faster for our low-bandwidth model. This result is not surprising, as the number of nodes in a supergraph increases more slowly as the block size $S$ decreases below $S = 512$ (see Table 1 above).

## B    SETTING THE VALIDATION ACCURACY TARGET

As explained in Section 4, we evaluate our approach on the *SeenProj VarMisuse* task of (Allamanis et al., 2018). For this task they report a test set accuracy of $85.5\%$ using their full model that includes node labels with learnable embeddings. In order to simplify the comparison and eliminate the influence of this additional learnable component, we turn off node labels in our implementation. Allamanis et al. (2018) report a test set accuracy of $84.3\%$ without node labels. However, these test set accuracies are achieved by a model trained on a larger training data than their published dataset; the difference are projects with a GPL licence that could not be released. In their Appendix D the authors report the test set accuracy of $84.0\%$ for their full model trained on the released data, i.e. $1.5$ percentage point lower than the model trained on the extended dataset. The test set accuracy using a model trained on the released data and without node labels is not provided, but we can estimate it to be approximately $83\%$ by combining the $1.5$ percentage point cost of only training on the released data, and the $1.2$ percentage point cost of turning off node labels.

When iterating on models and choosing hyper-parameters, it is important to use an independent validation dataset that does not pollute the test set. The dataset released by Allamanis et al. (2018) does come with a predefined validation split. Figure 4 shows the alignment of validation and test set accuracies, where we discovered that validation accuracies are well correlated with test set accuracies, but are generally 5 percentage points lower. The figure was produced by evaluating the validation and test set accuracies of models with different hyper-parameters and different randomness. We have explicitly not looked at which models achieved what test performance.

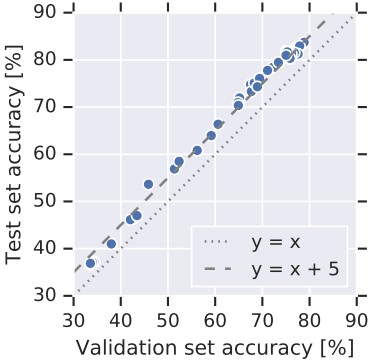 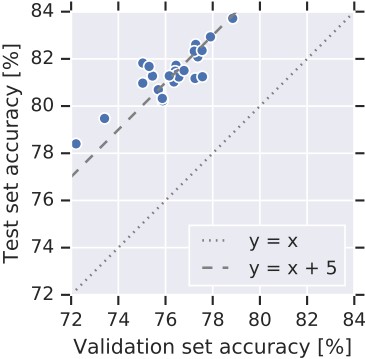

Figure 4:   Alignment of validation and test set accuracies on the *SeenProj VarMisuse* task of Allamanis et al. (2018). The plot on the right is a zoomed in version of the full plot on the left.

Combining the test set accuracy target of $83\%$ with the observation about validation accuracies being 5 percentage points lower, we set the target validation accuracy at $78\%$.

## C    WHAT ABOUT FILTERING GRAPHS?

In this section we consider the strategy of filtering out graphs that we cannot fully handle. In the context of our approach this means only training on training graphs whose bandwidth was reduced below the bandwidth limit that the model being trained is guaranteed to cover ($B \leq S - 1$, see Section 3.2). This filtering strategy is investigated in Sections C.1 and C.2 below.

We also investigate an alternative, simple approach based on filtering, that does not rely on bandwidths: dropping graphs with the largest numbers of nodes, so that for the graphs that remain it would be cheaper to have their adjacency matrices fully densified. See Section C.3 below.

## C.1 TRAINING SPEED ON BANDWIDTH-FILTERED DATASETS

Table 3 shows the training speed in graphs/second achieved for different combinations of model, hardware, and training data filtering criterion.

Table 3: Training speed (graphs/s) on slices of the training data with large bandwidth graphs discarded.

| graphs/second | GPU model | Low-bandwidth model | | | |
|---|---|---|---|---|---|
| | trained on V100 (16GB) | number of TPUv2 cores trained on | | | |
| | | 2 (16GB) | 8 (device) | 32 | 128 |
| Training data slice | | | | | |
| 100% (full data) | 80 | - | - | - | - |
| 91.9% ($B < 1024$) | 130 | 75 | 300 | 1,200 | 4,500 |
| 83.5% ($B < 512$) | 170 | 180 | 730 | 2,900 | 11,000 |
| 69.2% ($B < 256$) | 240 | 400 | 1,600 | 6,200 | 23,000 |
| 48.9% ($B < 128$) | 390 | 900 | 3,600 | 14,000 | 53,000 |

The bandwidth of a graph can be expected to be correlated with the number of nodes it has, so as the bandwidth requirement is strengthened the graphs being trained on tend to become smaller, and so more training graphs fit into one supergraph (see Table 4 below, and cf Table 1 above).

Table 4: Number of training graphs that fit into a *supergraph* on average. See Section C.1. The number of nodes in a *supergraph* is constant for the GPU model thanks to the sparse representation.

| Bandwidth bound | 127 | 255 | 511 | 1023 | None |
|---|---|---|---|---|---|
| Number of nodes $N$ in a *supergraph* (V100) | 110,000 | 110,000 | 110,000 | 110,000 | 110,000 |
| Average number of graphs in a *supergraph* | 370 | 230 | 170 | 130 | 70 |
| Number of nodes $N$ in a *supergraph* (TPUv2) | 70,272 | 65,536 | 49,125 | 28,672 | - |
| Average number of graphs in a *supergraph* | 240 | 140 | 70 | 30 | - |

## C.2 TIME TO VALIDATION ACCURACY ON BANDWIDTH-FILTERED DATASETS

Following the same procedure as in Section 5.3 where we trained on all training graphs (but ignoring some edges in large bandwidth graphs), we evaluated training time to 78% validation accuracy when training the GPU and low-bandwidth models on training data with large bandwidth graphs entirely discarded. We refer to these two settings as *lossy* and *filtered*, respectively. Table 5 reports the results. By comparing to the table in Figure 3, we can observe that while for block size $S = 1024$ on 8 and 32 TPUv2 cores it is more beneficial to *filter* out non-conforming graphs than to handle them in a *lossy* way, for block size $S = 512$ where the *lossy* approach led to the fastest training time to validation accuracy, the *filtering* approach does not scale nearly as well, and for $S = 256$ it is unable to reach the target validation accuracy at all.

## C.3 ELIMINATING GRAPHS BASED ON BANDWIDTH VS BASED ON NUMBER OF NODES

A simple strategy of training sparse models on dense hardware would be densification. Unfortunately, computing with dense representations of adjacency matrices is not feasible when some of the training graphs have a large number of nodes (in the (Allamanis et al., 2018) dataset the largest number of nodes in a single training graph is 20,231 in the released data, and 18,852 after the reachability-reduction described in Section 4). However, if one is content with throwing away some number of "cumbersome" training graphs anyway (such as those with large bandwidth), one can equally consider the natural alternative of throwing away graphs with many nodes. That way the remaining training graphs can have small enough adjacency matrices so that computations with fully densified representations become feasible.

In this section we investigate the question of how discarding graphs based on number of nodes compares to discarding based on bandwidth. To this end, for each bandwidth bound $\{1024, 512, 256, 128\}$ considered in this paper we computed the percentage of training graphs within

Table 5: Training steps (and training time) until $78\%$ validation accuracy on bandwidth-filtered data.

| | GPU model | Low-bandwidth model | | |
|---|---|---|---|---|
| Block size $S$ | trained on V100 | TPUv2 cores trained on | | |
| | | 8 | 32 | 128 |
| all edges | 75,000 (19 h) | - | - | - |
| 1024 | 115,000 (33 h) | 12,000 (3 h) | 2,500 (32 min) | 2,200 (30 min) |
| 512 | 115,000 (38 h) | 18,000 (4 h) | 14,000 (3 h) | 10,500 (2.5 h) |
| 256 | not reached | not reached | not reached | not reached |
| 128 | not reached | not reached | not reached | not reached |

the bound (already reported in the table in Figure 1 (middle)) and then found a corresponding bound on the number of nodes that leads to the same percentage of conforming training graphs. For example, $83.5\%$ training graphs have a bandwidth smaller than $512$, and if we choose a bound of $2037$ on the number of nodes, $83.5\%$ training graphs will have a smaller number of nodes than this bound. See the first two columns of Table 6. We then trained the (sparse) GPU model on training datasets filtered based on these bandwidth and number of nodes bounds, and evaluated the validation accuracy reached at specific training steps. In each case we perfored a random hyper-parameter search over the ranges reported in Appendix E, using $40$ hyper-parameter configuration samples. Columns 3-5 of Table 6 report the best obtained results for each case.

While at $100\,000$ training steps filtering based on number of nodes can in some cases lead to slightly higher validation accuracies, in all other cases filtering based on bandwidth appears to works better. In particular, at $300\,000$ training steps all training runs seem to have converged, and filtering based on bandwidth always lead to higher final validation accuracy.

Filtering based on number of nodes versus based on bandwidth has different computational implications for dense hardware implementations. While densifying (and pre-multiplying by) an $N$-node adjacency matrix incurs a cost of $N^2$, our low-bandwidth message passing algorithm (Section 3.2 for graphs with $N$ nodes and bandwidth $B$ incurs a cost of (approximately) $3NB$. The last column of Table 6 reports the estimated per-node computational cost of densification under the appropriate model. One unit of cost corresponds to densifying $1000$ matrix elements per graph node. We can see that in all cases, under a fixed proportion of graphs that can be thrown away, filtering based on bandwidth not only tends to lead to higher validation accuracies, but does so with lower computational cost.

Table 6: Validation accuracy on varying slices of the training data.

| Training data slice | | Validation accuracy at training step | | | Iteration |
|---|---|---|---|---|---|
| size | definition | $100\,000$ | $200\,000$ | $300\,000$ | cost |
| $100\%$ | all data | 78.1 | 78.5 | 78.9 | 18.6 |
| $91.9\%$ | bandwidth $< 1024$ | 77.9 | **78.5** | **78.9** | 3.1 |
| $91.9\%$ | order $< 3493$ | **78.2** | 78.3 | 78.4 | 3.5 |
| $83.5\%$ | bandwidth $< 512$ | 78.0 | **78.2** | **78.4** | 1.5 |
| $83.5\%$ | order $< 2037$ | **78.1** | 78.1 | 77.5 | 2.0 |
| $69.2\%$ | bandwidth $< 256$ | **77.5** | **77.7** | **77.8** | 0.8 |
| $69.2\%$ | order $< 1191$ | 76.6 | 76.0 | 75.8 | 1.2 |
| $48.9\%$ | bandwidth $< 128$ | **74.8** | **74.6** | **74.7** | 0.4 |
| $48.9\%$ | order $< 624$ | 73.9 | 73.6 | 73.7 | 0.6 |

*Remark.* For the full training data the computation cost has been calculated as $\max(3 \times 9\,372, 18\,582) = 18\,582$, where $9\,372$ and $18\,582$ are the largest bandwidth and number of nodes in the (reachability-reduced, see Section 4) training data, respectively.

To conclude this section, we note one conceptual advantage of handling graphs with low-bandwidth rather than with a low number of nodes. When multiple low bandwidth graphs are placed into a *supergraph* using the *sparse batching* technique described in Section 2, the resulting supergraph is automatically low-bandwidth and our low-bandwidth message passing method applies. On the other hand, when multiple graphs with a low number of nodes are placed into a *supergraph*, this supergraph would have a large number of nodes (sum of node counts of the individual graphs) and so either one again needs a custom densification scheme matching the sparsity structure of the *supergraphs*'s adjacency matrix, or a different batching strategy needs to be applied.

## D    Low-bandwidth model on GPU

Out of interest it is also possible to train the low-bandwidth model on GPU. We can confirm that the low-bandwidth model, designed with dense hardware in mind, is slower on a single V100 GPU than the highly optimized sparse GPU implementation used in the paper. The speeds were as follows (compare to Fig. 2): 7 graphs/s ($S = 1024$), 16 graphs/s ($S = 512$), 31 graphs/s ($S = 256$) and 57 graphs/s ($S = 128$).

## E    Details for reproducibility

**Reachability preprocessing**    As a one-time data preprocessing step we discarded nodes from all graphs that cannot affect model predictions (and so do not influence training). Specifically, following Allamanis et al. (2018) we use $T = 8$ message passing steps in the GGNN encoder, so only nodes from which a candidate node is reachable within 8 steps affect the final embeddings and thus the logits of these candidates. Any other nodes can be discarded. All reported results including the GPU baselines use reachability-reduced data. Reachability preprocessing was a final tweak that gave a small speed-up to all methods; without it the results were similar and the conclusions identical.

**Batch size**    All GPU training runs used a *supergraph* with $110\,000$ nodes. The number of nodes in a *supergraph* for the low-bandwidth model trained was given in Table 1. When training on multiple cores, each core independently received a *supergraph* with this many nodes, i.e. the total number of nodes used in a single gradient update scaled linearly with the number of cores in the system.

**Model**    All our models used $H = 128$ hidden dimensions, $T = 8$ message propagation steps, and a learnable per-edge-type bias term in the message propagation. We did *not* use message averaging, i.e. messages incoming into a node were always summed, not averaged.

**Optimization**    All our training runs used the Momentum optimizer. Further optimization techniques that we employed were *dropout* (on the GRU cell), *label smoothing*, *weight decay* ($L_2$-regularization), a *linear learning rate decay* schedule, *Nesterov* momentum, and *gradient clipping* (by norm).

All GPU training runs were started with 40 random hyper-parameter configurations independently sampled from the following values:

- `dropout_keep_prob` $\in \{0.6, 0.7, 0.8, 0.9\}$
- `label_smoothing` $\in \{0, 0.005, 0.05\}$
- `weight_decay` $\in \{10^{-1}, 10^{-6}, 10^{-5}, 10^{-4}, 10^{-3}\}$
- `learning_rate` $\in \{0.03, 0.1, 0.3, 1.0, 3.0\}$
- `learning_rate_decay_steps` $\in \{100000, 300000\}$
- `end_learning_rate_factor` $\in \{0.1, 1.0\}$
- `momentum` $\in \{0.3, 0.5, 0.7, 0.9, 0.95, 0.97\}$

- `use_nesterov` ∈ { `False`, `True` }
- `gradient_clip` ∈ $\{0.003, 0.01, 0.1, 0.3\}$

Note the random search automatically marginalizes out any potentially irrelevant hyper-parameters; for example, when `end_learning_rate_factor` = 1.0, the value of `learning_rate_decay_steps` does not matter.

Training runs of our low-bandwidth model on TPUv2 were also started with $40$ independently sampled random hyper-parameter configurations. As the batch size increased, we allowed the TPUv2 model to use larger learning rates and a smaller number of learning rate decay steps:

- `learning_rate` ∈ $\{1.0, 3.0, 7.0, 10.0, 30.0\}$
- `learning_rate_decay_steps` ∈ $\{500, 1000, 5000, 10000\}$

We had checked that these options would not have aided the GPU baseline.

