# OpenReview forum: "Fast Training of Sparse Graph Neural Networks on Dense Hardware"
_ICLR.cc/2020/Conference — Reject_

### Official Review · AnonReviewer3 · 2019-10-16
**Official Blind Review #3**

**Rating:** 6

**Review:**

The authors present a framework to implement graph neural networks training
efficiently ---an inherently sparse task--- using "custom dense hardware", here
Tensor Processing Units (TPUs V2). The key steps are: (1) reordering the labels
to reduce the bandwidth of the adjacency matrix, (2) (Sometimes approximate)
Decomposition using block matrices for efficient storage and computations, and
(3) memory layout optimization. They evaluate their framework on the VarMisuse
dataset and compare the performance against a GPU implementation running on
Nvidia Tesla V100. In the best configuration, they were able to reach 78% test
accuracy in 13 minutes vs 19h for the baseline.

The paper is well structured and thorough. I was able to understand the
challenges and the solutions proposed, even without prior knowledge in the
architecture this work focuses on. I think it is sufficiently detailed to enable
an independent implementation without referring to the authors source code.

However, I feel that it might lack novelty. Indeed, as described in the Related
Work section, each component of the pipeline is well known and used very
frequently in the HPC community. Sometimes, knowing how arrange common
primitives is very powerful, and looking at the results of the experimental
section, this is enough to improve performance by almost two orders of
magnitude. However, I think it might be more due to some intrinsic properties of
the dataset than the method itself. As made clear by the title of section 5.1,
it is the data itself that has low bandwidth. If we consider a dataset that does
not satisfy this requirement, stage (1) has no effect, and it is impossible to
perform the decomposition done in (2). In that situation, the contributions of
this paper would be nullified.  It would be perfectly reasonable to think that
most datasets have low bandwidth, but this not a claim that made the authors.
This work would be considerably more impactful if it measured the bandwidth of
more well recognized datasets. My small exposition to these problem does not
allow me to make any meaningful suggestion.

Finally, I am puzzled by the "dense hardware"/GPU distinction. From my
experience, GPU devices *are* designed for and extremely efficient at dense linear
algebra. Sparse operations are historically performed on CPUs. While they are
possible on GPUs, they are usually much slower. For example CUSparse, the sparse
matrices library, part of NVidia CUDA toolkit, was only introduced in its
version 4. It's a clear indication that sparse operations are not a strength of
GPUs. According to my experience writing GPU code, I feel that this approach
would actually perform extremely well on GPUs as it does on TPUs. I think it is
thus important to compare this framework on GPUs too. Since these
optimizations are not TPU specific and have not been applied in the GPU based
GNN libraries referenced in this paper reinforce my concerns that they are
problem-specific.

Even though the performance gains demonstrated are sizeable, the fact the
approach does not seem TPU specific and is potentially problem specific makes me
lean towards rejection.


**Experience Assessment:**

I have read many papers in this area.

**Review Assessment: Checking Correctness Of Derivations And Theory:**

N/A

**Review Assessment: Checking Correctness Of Experiments:**

I assessed the sensibility of the experiments.

**Review Assessment: Thoroughness In Paper Reading:**

I read the paper thoroughly.

---

> ### Author Response · Authors · 2019-11-10
> **Author Response**
>
> Thank you for reviewing our work, and for praising the clarity and detail of our presentation. We'd like to start by addressing the review's final two paragraphs first, as we think we can clear up the puzzlement.
>
> > Paragraph starting "Finally, I am puzzled by the "dense hardware"/GPU distinction. "
>
> As we tried to explain in paragraph 3 of the Introduction, GPUs certainly perform well on dense operations, but they are flexible enough to efficiently support sparse operations. The "dense hardware" that we are targeting is even more specialized to dense operations than GPUs. For example, hardware based on systolic arrays (https://en.wikipedia.org/wiki/Systolic_array ) can lead to even better dense matrix-multiply performance, but at the cost of flexibility in supporting sparse operations. Note that TPUs are mentioned as a prominent example of this kind of hardware in the above wikipedia link.
>
> We acknowledge that "dense hardware" is not a standardized term, and we're open to using alternative terminology that would be clearer if any of the reviewers have suggestions.
>
> > It's a clear indication that sparse operations are not a strength of GPUs.
>
> While we agree GPUs may not be the optimal hardware for sparse operations, our experience is that it is better to place sparse graph neural network operations on GPU than CPU. It is possible to achieve good GPU utilization and also avoids having to copy activations back and forth from CPU to GPU during GNN propagation.
>
> > According to my experience writing GPU code, I feel that this approach would actually perform extremely well on GPUs as it does on TPUs. I think it is thus important to compare this framework on GPUs too.
>
> Actually, this experiment is included with the submission in Appendix D. We found that training was slower on a single V100 GPU than on a TPU with the same amount of RAM, which gives additional support to our above claims that what we're calling "dense hardware" is more specialized to dense operations than GPUs.
>
> > Even though the performance gains demonstrated are sizeable, the fact the approach does not seem TPU specific and is potentially problem specific makes me lean towards rejection.
>
> In light of the above, we'd ask the reviewer to reconsider their opinion on this. While we agree the approach can be run on GPUs, the benefit only comes when running on hardware that is more specialized to dense (as opposed to sparse) operations than GPUs. If we understand correctly, this addresses a major concern of the review. If not, could you please help us understand what we're still missing?

---

> > ### Comment · AnonReviewer3 · 2019-11-11
> > **Reviewer #3 answer**
> >
> > Thanks for the clarification, this is very helpful.
> >
> > I indeed missed appendix D, the text based presentation made it hard to read. One last question, the performance there seems off  though, with only 128 edge is still 25% slower than than full sparse implementation on the same hardware. Any insight on why is that, this seem extremely surprising.

---

> > > ### Author Response · Authors · 2019-11-11
> > > **Author Response**
> > >
> > > Even at the smallest block sizes (e.g., 128), the dense implementation requires significantly more FLOPS than the sparse implementation.
> > > In more detail, we use the symbol S to denote the block size (as described in Section 3.2), so one propagation step of the low-bandwidth model is O(N*S), where N is the number of nodes. For the sparse model, one propagation step is O(E), where E is the number of edges. Since the graphs have less than 10 edges per node on average, 10*N << N*S, which can explain why the sparse model is faster in this setting on GPU. The wall-clock performance comes down to how specialized the hardware is to dense operations (i.e., how much faster the hardware can execute dense versus sparse operations). Please let us know if we misunderstood your question, or if this clarifies things.

---

> > > ### Author Response · Authors · 2019-11-14
> > > **Author Question**
> > >
> > > We were just wondering if you have any other questions before the time window for us to respond closes, or if the assessment of our work has shifted in any way. If so, we'd appreciate it if you'd update your score. Thanks.

---

### Official Review · AnonReviewer1 · 2019-10-18
**Official Blind Review #1**

**Rating:** 3

**Review:**

The authors propose a method to speed-up the time to validation accuracy for a particular class of graph neural networks:  Gated graph sequence neural networks (GGSNNs).

The paper is interesting in that it describes several operations and engineering considerations to speed up the processing of a GGSNN on TPUs. It is essentially a collection of engineering steps that improve the time to validation accuracy.

While I'm not an expert in (G)NN acceleration on TPUs, I have experience with GNNs and approaches to accelerate CNNs in GPUs. My assessment is that the scope of this work is far too narrow. It is specific to GGSNNs which is a small family of GNNs not widely used. It is also specific to TPUs and lacks evaluations of the proposed approach on other type of hardware.

It is for these reasons that I think the paper is not appropriate for ICLR. The scope has to be broadened both in terms of the NN models and the hardware types.


**Experience Assessment:**

I have read many papers in this area.

**Review Assessment: Checking Correctness Of Derivations And Theory:**

I assessed the sensibility of the derivations and theory.

**Review Assessment: Checking Correctness Of Experiments:**

I assessed the sensibility of the experiments.

**Review Assessment: Thoroughness In Paper Reading:**

I read the paper at least twice and used my best judgement in assessing the paper.

---

> ### Author Response · Authors · 2019-11-09
> **Author Response**
>
> Thank you for taking the time to review our work. We would like to respond to both raised concerns: the method being (1) specific to a model family that is “not widely used”, and (2) “specific to TPUs”.
>
> (1) We would like to point out that the claim in the review that our work is specific to “Gated graph sequence neural networks (GGSNNs)” is imprecise. The confusion may have arisen because the cited paper [Li et al., 2016] introduces both Gated Graph Neural Networks (GGNNs) as well as GGSNNs, but as explained and described at the beginning of Section 2, we use GGNNs to demonstrate our method.
>
> We are not sure how to best define “widely used”, but GGNNs have been successfully used not only for program understanding understanding and generation [Allamanis et al., ICLR 2018; Brockschmidt et al., ICLR 2019], but also in models with graph-structured external memory [Johnson, ICLR 2017], in computer vision [Marino et al., CVPR 2017; Li et al., ICCV 2017; Chuang et al., CVPR 2018], music generation [Jeong et al., ICML 2019], and molecule generation [Liu et al., NeurIPS 2018]. This is just a subset of uses that we were able to find from a brief Google scholar search; the paper introducing it has now accumulated over 600 citations and has been applied broadly.
>
> Further, we focus on the GGNN variant because that was the variant used in the work whose dataset and model we are building on [Allamanis et al., ICLR 2018], but the techniques that we present could straightforwardly be adapted to other graph neural network variants, such as those categorized as "message passing neural networks" by Gilmer et al. [ICML 2017].
>
> (2) Apart from the GPU evaluation in Appendix D, we have indeed only evaluated our method on TPUs, which we believe to be representative of the class of “dense hardware” (for lack of a better term) as described in the 3rd paragraph of the Introduction. However:
> a) The claim that the method is specific to TPUs is imprecise — by having eliminated the need for any sparse operations, it makes training sparse GGNNs possible on any hardware where sparse operations are unavailable or slow.
> b) Our contributions 1-3, including empirical results on large-batch training for GNNs, and the observation that GNN training is robust to dropping some non-conforming edges during training time, are entirely independent of TPUs and the results would be identical on any hardware (that permits fast enough training to actually compute these results).
>
> Please let us know your thoughts (and any further reconsideration would be appreciated).

---

> > ### Comment · AnonReviewer1 · 2019-11-09
> > **Response**
> >
> > Thanks for engaging in a discussion. I’ll later respond more in detail to your clarifications but let me ask you a few quick questions:
> >
> > If, as you claim, the contributions of your work also allow you, without much effort, to improve the performance for message-passing NN type models such as GCNs whose use is undoubtedly wide, why not also run these experiments?
> >
> > If, as you claim, it is also possible to train sparse GNNs in other hardware than TPUs (which are used much less widely than GPUs), what is the reason for not running these experiments?
> >
> > I’m really trying to understand this since it would make your paper so much stronger and so much more relevant to a lot more people.
> >
> > It was rather the background section (in conjunction with the intro) that might have confused me. In the background section you introduce a very specific GNN while in the intro you talk about GNNs in general. So let me ask you this:
> > The contributions that you are presenting here, are they also applicable to other types of GNNs and how do we know that we’ll see an improvement there (either in theory or through experiments)?
> >
> > I’m happy to continue a discussion here. My initial assessment might have been indeed imprecise.
> >
> > Finally, and this is more of a comment to the AC, I’m not an expert in accelerating ML models with TPUs and consider that niche.  Perhaps I’m wrong here as well.

---

> > > ### Author Response · Authors · 2019-11-10
> > > **Response**
> > >
> > > Thank you for engaging in a constructive manner and for your willingness to continue the discussion.
> > >
> > > There seem to be two parts to the question about our choice of GNN variant:
> > > (1) Why did we choose GGNNs instead of GCNs?
> > > (2) Why didn't we run more experiments with more GNN variants?
> > >
> > > For (1), we actually didn't start by thinking about which GNN variant we wanted to work with. We started by thinking about which application we wanted to address. We arrived at the Variable Misuse task for the following reasons:
> > > (a) It was an oral presentation at ICLR 2018, which indicates it's a problem considered important by the ICLR/ICML/NeurIPS community.
> > > (b) The graphs are large and sparse.
> > > (c) Training times are reasonably long, so there would be benefit in speeding them up. We also imagine that in future work the program source code data sets could be scaled up significantly.
> > > (d) The graphs don't obviously have tractable structure in them. There's an interesting mix to the graph where there is a superposition of sequential structure (due to the sequence of tokens in source code), tree structure (due to the abstract syntax tree), and longer-range edges (e.g., connecting uses of the same variable across the code).
> > >
> > > Having decided that Variable Misuse was a good application to target, we followed the design choices from the state of the art approach to the problem [Allamanis et al., 2018].
> > >
> > > For (2), the reason is simply that large-scale performance benchmarking experiments are expensive to run carefully, both in terms of machine and human time. Given a finite budget of both, we decided it would be more compelling to put energy into carefully designing experiments and running ablations on one task than spreading our efforts more thinly across a variety of tasks. Similar reasoning applies to our decision to run on GPUs and TPUs, rather than trying to find and spread efforts across other hardware platforms.
> > > We believe our choice of experimental protocols achieves two key goals: (a) convincingly shows that there exist sparse graph neural network models that can be trained fast on dense hardware, and (b) provides useful guidance to future researchers and practitioners who would like to speed up related workloads on related hardware.
> > >
> > > We hope that readers can see that we haven't over-optimized to the Variable Misuse task or GGNN variant. We discovered that approximately low bandwidth structure can be revealed in the [Allamanis et al., 2018] dataset, and we have made it clear in the paper that this is the crucial assumption about the data. However, we are optimistic that there are other datasets with approximately low bandwidth structure, and/or approximately some other structure that could be leveraged in an analogous way. While it would be more work to adapt our work to other structures, we hope that our paper inspires GNN practitioners to look for this structure, even if it may only approximately hold. We would like to think that a practitioner is significantly better off addressing the next problem after having read our paper than if they hadn't.
> > >
> > > For the question of GNN variants, the reason why we believe the method applies to other variants is the following:
> > > * Dense operations are relatively straightforward to handle, and this is where most of the variation in the alternative MPNN variants comes from.
> > > * The key challenge is the sparse operations that aggregate messages from neighbors in the sparse graph. This corresponds to Eq 1 of https://arxiv.org/abs/1704.01212 . Eq 3 in our submission generically applies.
> > >
> > > The specific algorithm (Algorithm 1) would need to be modified, and perhaps the memory layout should be reconsidered, but the core idea is generally applicable.
> > >
> > >
> > > Is this a niche topic?
> > >
> > > We'd ask the reviewer to consider a slightly broader view of the work. While it's true that we focus our energies on a specific model and application, we believe there are broader implications where our work provides insight (though we only claim that it is part of the discussion; not the final word). For example, consider the (very broad) question of what hardware the community should build for deep learning in the future. One question is if special hardware should be built to support sparsity, or if it suffices to double-down on "dense hardware". Prior to our work, one might think that doubling down on dense hardware means we'd lose support for sparse graph neural networks. Our work provides one data point showing that this isn't necessarily the case, and that the issue is more subtle.

---

> > > ### Author Response · Authors · 2019-11-14
> > > **Additional response to the GCN question**
> > >
> > > An additional note on the applicability of our method on GCNs: at its core, our method targets speeding up the multiplication of the sparse adjacency matrix A of size [N x N] with a dense node representation matrix E of size [N x H] with N being the number of nodes and H being the dimensionality of the node representations.  Most GNN variants have this sparse-dense matrix multiplication as (at least one of) the key operation, including GCNs, which compute the product AXW = A(XW) at each layer, where X is a node representation matrix, W is the weight matrix for a linear transformation.  XW is a dense matrix multiplication, but computing the product of A and XW is the more expensive sparse-dense matrix multiplication, which our algorithm would apply equally well.
> > >
> > > We were wondering if you have additional questions, and/or if your assessment of the work has shifted in light of our discussions? If so, we'd appreciate it if you'd update your score. Thanks.

---

> > > > ### Comment · AnonReviewer1 · 2019-11-15
> > > > **Response to rebuttal**
> > > >
> > > > Thank for providing additional information that clarifies the scope and possible applicability of the proposed approach.
> > > >
> > > > I increase my score to a "weak reject" only, however, because I still think your submission is falling short in several ways.
> > > >
> > > > Based on your paper and what you wrote in the rebuttal, the title of the paper should be "Fast Training of GGNNs on TPUS." Your title and intro are written in a general way as if you already have results about other GNNs and other sparse hardware.  The explanations as to why the approach could also work for other GNNs and on other hardware could be in a discussion section. To make an analogy: it is like engineering changes for a particular Resnet architecture on TPUs and then calling the paper "Fast Training of CNNs on Sparse Hardware." Of course one can argue that due to certain common components (pooling, convolutions, etc.) of CNNs other architectures might also benefit. As someone who has worked on accelerating CNNs on GPUs, I know though that different CNNs can behave still very differently.
> > > >
> > > > Narrowing down the scope in the intro (and other parts of the paper) and make good arguments *in the paper* as to why this is an important problem and why some results might generalize to other problems, would be a good step to improving the paper. Of course, it would be even better if you would also evaluate your ideas on other problems, GNNs, hardware.
> > > >
> > > > While I agree that the problem in [Allamanis et al., 2018] is interesting and has the "right" types of graphs to show what you set out to show, I disagree with the reasoning"it was an oral, therefore it's an important problem, therefore it's enough to focus on that problem to show gains for GNN acceleration."

---

> > > > > ### Author Response · Authors · 2019-11-15
> > > > > **Response**
> > > > >
> > > > > Thanks for your response! We would like to respond to the persisting concerns about applicability to other GNN variants and the hardware evaluated on.
> > > > >
> > > > > (1) Applicability to other GNN variants
> > > > >
> > > > > We believe our situation is qualitatively different from the example of CNN acceleration on GPUs, because with dense hardware replacing a sparse operation with dense ones is a substantially larger jump than is the scale of any speed variations arising due to different dense operations appearing in different GNN variants.  A probably closer analogy to CNNs would be speeding up the convolution operation which is used by all CNN variants (ResNet, Inception, AlexNet etc.), rather than optimizing for a particular architecture.
> > > > >
> > > > > To confirm this is really the case and that our method is applicable to other GNN variants, we’ve run a last minute experiment measuring the training speed of the Graph Convolutional Network (GCN) [Kipf and Welling, 2017] variant, keeping all other configurations the same. We were able to run on a single TPUv2 device (8 cores) and thus provide a direct comparison to the numbers reported in Figure 2 in the submission for the GGNN variant:
> > > > >
> > > > > graphs/sec | GGNN | GCN
> > > > > S=1024 | 100 | 110
> > > > > S=512 | 260 | 280
> > > > > S=256 | 450 | 500
> > > > > S=128 | 630 | 710
> > > > >
> > > > > Training a GCN turns out to be in fact slightly faster than training the GGNN variant, presumably because the GCN-specific dense operations (symmetric normalization, sigmoid) are slightly cheaper than the GRU step in the GGNN.
> > > > >
> > > > > (2) Hardware evaluated on
> > > > >
> > > > > We are happy to make it clearer from the outset that in this work, from the class of dense hardware we only evaluate on TPUs. However, we still think that the highest value lies in showing that using dense hardware does not necessarily mean losing the ability to train sparse GNNs (on the contrary, it can lead to fast training speeds), and for such an existence claim a demonstration on one class of dense hardware already brings the largest added value.
> > > > >
> > > > > Any further reconsideration would be appreciated.

---

### Official Review · AnonReviewer2 · 2019-10-23
**Official Blind Review #2**

**Rating:** 6

**Review:**

This paper proposes a method to train graph neural networks on dense hardware such as TPUs. The method is motivated by an observation that connections in graphs have locality in some datasets.  Experiments show significant improvements in training speed compared to single-GPU training.

The overall score of this paper is slightly positive. There is a certain demand to perform training on hardware targeted to dense computations. Even though the applications of the proposed method is limited to data with low-bandwidth, the paper shows there are real applications of the method. The effectiveness of the proposed method is well-supported by the experiments.

Major comments:
Comparisons with single-GPU training can be unfair. The method in Ma et al. (2018) is indeed not easy to scale many GPUs because their target is processing extremely large graphs in parallel. Since the experiments in the submitted paper use relatively small graphs that fit in a single GPU memory, it will not be so challenging to scale many GPUs. At least, it is recommended to compare the results with training on several GPUs using data-parallel execution implemented in TensorFlow (or any other suitable frameworks). If it is difficult, please provide more specific reasons why it is challenging to perform multi-GPU training.

**Experience Assessment:**

I do not know much about this area.

**Review Assessment: Checking Correctness Of Derivations And Theory:**

I assessed the sensibility of the derivations and theory.

**Review Assessment: Checking Correctness Of Experiments:**

I assessed the sensibility of the experiments.

**Review Assessment: Thoroughness In Paper Reading:**

I read the paper at least twice and used my best judgement in assessing the paper.

---

> ### Author Response · Authors · 2019-11-14
> **Author Response**
>
> Thanks for your encouraging review!
>
> We don't claim that it is impossible to match our training speeds using a large number of GPUs, but we are not aware of any work that has successfully done so. Our claim in this regard is simply that we have achieved training speeds that are far better than any existing results. While we agree Ma et al. [2018] focus on larger graphs, we do not think all the challenges they encounter could be totally avoided on the Allamanis et al. [2018] dataset that we use. For example, we believe the challenges related to shared PCIe interconnect [Ma et al. 2018, Sec. 6.3] would still persist.
>
> We reported single GPU training times to establish that training the model to state of the art accuracy takes a reasonable amount of time. This helps contextualize the results we get on multi-TPU training.
>
> We'd like to reiterate that before this work, it was not clear that it would be possible to use dense hardware to train sparse GNNs in any reasonable timeframe at all, because the hardware is very specialized to fixed-size dense computation, and a naive densification of large graphs isn't feasible. Our results showing that it's possible are valuable because this style of dense hardware is becoming increasingly prevalent as hardware becomes more specialized to matrix multiply-based workloads. Since the presented techniques did allow us to train on TPUs, we exploited the ease of scaling up to 512 cores (with TPUs it is a matter of changing a single parameter) in order to report results from large-batch training of sparse GGNNs, and we also pointed out the fast training time one can achieve this way.

---

### Author Response · Authors · 2019-11-15
**Author Response after discussion**

We thank the reviewers for engaging with us in constructive discussions. The discussions have been productive, and our clarifications about why "dense hardware" is different from GPUs and why we chose to focus on the task from Allamanis et al (2018) have led to reviewers increasing their scores, with two of the three reviewers now favoring acceptance.

The remaining sticking points are summarized in Reviewer1's most recent response ( https://openreview.net/forum?id=BklXkCNYDB&noteId=SJegjtkhsB ). Due to the timing Reviewer 1 may not have yet had a chance to see our latest response, but we are summarizing the discussion as it stands due to our time window for responding closing. The issues appear to come down to the generality with which the title and introduction are written:
"it is like engineering changes for a particular Resnet architecture on TPUs and then calling the paper "Fast Training of CNNs on Sparse Hardware."" (Reviewer1)

First, we think this is a helpful phrasing of the issue, because it highlights how we see things differently from the reviewer. Specifically, we don't think this analogy is accurate because prior to our work, there was no practical way to train sparse graph neural networks on any dense hardware when there were large graphs in the dataset. Our key contribution is an existence proof showing that there is a configuration of (GNN variant, hardware, dataset) where fast training is possible. In the hypothetical situation posed by the reviewer, previous work would have already established that fast training of computer vision architectures on TPUs was possible, so an existence proof would not be a contribution.

Second, we have now further supported the claim that it is straightforward to adapt the ideas in the paper to other GNN variants by implementing the GCN variant from Kipf & Welling [2017]. Results appear in our most recent response to Reviewer1 ( https://openreview.net/forum?id=BklXkCNYDB&noteId=SkgsqyQnjS ). They support our claim that the ideas presented in the submission apply to other kinds of GNNs.

Finally, we acknowledge that there is subtlety in our claims related to kinds of hardware, and we will take a careful pass through the writing to make sure we are very precise in our claims:
* We have developed the algorithm by considering an abstraction ("dense hardware") where matrix multiplies are very fast and sparse operations are very slow, and then developed an algorithm that is tailored to these assumptions.
* We have shown experimentally that there exists a (GNN variant, hardware, dataset) configuration where sparse graph neural networks with large graphs can be trained quickly. Previously, there was no configuration where this was practical.
* The only dense hardware that we have run experiments with is TPUs.
(If the use of the term "dense hardware" in the title versus "TPU" is a major sticking point, then we're willing to change it.)

---

### Decision · Program_Chairs · 2019-12-19

**Decision:**

Reject

**Comment:**

While there was some interest in the ideas presented, this paper was on the borderline, and was ultimately not able to be accepted for publication at ICLR.

Reviewers raised concerns as to the novelty, generality, and practicality of the approach, which could have been better demonstrated via experiments.